

# Development and validation of a novel risk score to predict 5-year mortality in patients with acute myocardial infarction in China: a retrospective study

Yan Tang[1,*], Yuanyuan Bai[1,*], Yuanyuan Chen[1], Xuejing Sun[1], Yunmin Shi[1], Tian He[1], Mengqing Jiang[1], Yujie Wang[1], Mingxing Wu[2], Zhiliu Peng[2], Suzhen Liu[1], Weihong Jiang[1], Yao Lu[3], Hong Yuan[1,3] and Jingjing Cai[1,3]

[1] Department of Cardiology, The Third Xiangya Hospital, Central South University, Changsha, Hunan, China
[2] Department of Cardiology, Xiangtan Central Hospital, Xiangtan, Hunan, China
[3] Center of Clinical Pharmacology, The Third Xiangya Hospital of Central South University, Changsha, Hunan, China
* These authors contributed equally to this work.

Corresponding author
Jingjing Cai,
caijingjing83@hotmail.com

## ABSTRACT

**Background:** The disease burden from ischaemic heart disease remains heavy in the Chinese population. Traditional risk scores for estimating long-term mortality in patients with acute myocardial infarction (AMI) have been developed without sufficiently considering advances in interventional procedures and medication. The goal of this study was to develop a risk score comprising clinical parameters and intervention advances at hospital admission to assess 5-year mortality in AMI patients in a Chinese population.

**Methods:** We performed a retrospective observational study on 2,722 AMI patients between January 2013 and December 2017. Of these patients, 1,471 patients from Changsha city, Hunan Province, China were assigned to the development cohort, and 1,251 patients from Xiangtan city, Hunan Province, China, were assigned to the validation cohort. Forty-five candidate variables assessed at admission were screened using least absolute shrinkage and selection operator, stepwise backward regression, and Cox regression methods to construct the C2ABS2-GLPK score, which was graded and stratified using a nomogram and X-tile. The score was internally and externally validated. The C-statistic and Hosmer-Lemeshow test were used to assess discrimination and calibration, respectively.

**Results:** From the 45 candidate variables obtained at admission, 10 potential predictors, namely, including **C**reatinine, experience of **C**ardiac arrest, **A**ge, N-terminal Pro-**B**rain Natriuretic Peptide, a history of **S**troke, **S**tatins therapy, fasting blood **G**lucose, **L**eft ventricular end-diastolic diameter, **P**ercutaneous coronary intervention and **K**illip classification were identified as having a close association with 5-year mortality in patients with AMI and collectively termed the C2ABS2-GLPK score. The score had good discrimination (C-statistic = 0.811, 95% confidence intervals (CI) [0.786–0.836]) and calibration (calibration slope = 0.988) in the development cohort. In the external validation cohort, the score performed well in both discrimination (C-statistic = 0.787, 95% CI [0.756–0.818]) and calibration

(calibration slope = 0.976). The patients were stratified into low- (≤148), medium- (149 to 218) and high-risk (≥219) categories according to the C2ABS2-GLPK score. The predictive performance of the score was also validated in all subpopulations of both cohorts.

**Conclusion:** The C2ABS2-GLPK score is a Chinese population-based risk assessment tool to predict 5-year mortality in AMI patients based on 10 variables that are routinely assessed at admission. This score can assist physicians in stratifying high-risk patients and optimizing emergency medical interventions to improve long-term survival in patients with AMI.

## INTRODUCTION

Acute myocardial infarction (AMI) is the leading cause of mortality and morbidity worldwide. However, the mortality rate attributed to AMI remains variable across different countries (*Khan et al., 2010*; *Fox et al., 2010*; *Roth et al., 2020*). With the introduction of new treatments and assessment tools over the past several decades, the overall mortality of AMI has been declining in the Western countries (*Rogers et al., 2008*; *Yeh et al., 2010*; *Laribi et al., 2014*; *Reed, Rossi & Cannon, 2017*), However, in China, the long-term mortality of AMI has not significantly decreased, and the 5-year mortality rate is approximately 20% of 5-year mortality (*Fox et al., 2010*; *Khan et al., 2010*; *Yeh et al., 2010*; *Li et al., 2015*; *Chang, Liu & Sun, 2017*; *Szummer, Jernberg & Wallentin, 2019*), even though the 1-year mortality of AMI patients in China has markedly decreased from approximately 22% in 1995 to 11% by 2014 (*Fox et al., 2010*; *Roth et al., 2020*). Moreover, by 2030, the number of AMI patients in China is expected to reach 23 million which might translate to higher long-term mortality (*Song et al., 2020*).

The Global Registry of Acute Coronary Events (GRACE) score is a widely used tool in clinical practice for the prediction of mortality for up to 5 years in patients with acute coronary syndrome (ACS) (*Kozieradzka et al., 2011*; *Zdanyte et al., 2020*). However, the GRACE score was established based on clinical parameters at hospital admission, without consideration of revascularization intervention and medication advances in recent years. Furthermore, contemporary risk prediction scores such as the Korea Acute Myocardial Infarction Registry (KAMIR) score based on the Korean population have mainly focused on the factors assessed at hospital discharge, which may not be beneficial for optimizing emergency care and in-hospital treatments for patients with AMI (*Kim et al., 2011*).

Therefore, we developed and validated a risk assessment score that integrates clinical variables at hospital admission and advances in interventional procedures and medication to predict 5-year mortality in patients with AMI. This tool will assist physicians in identifying patients who are at high risk of long-term mortality and thus optimize the medical treatments to improve the survival of AMI patients in China.

## MATERIALS AND METHODS

### Study populations and procedure

We conducted a retrospective observational study of 3,088 inpatients with International Classification of Disease (ICD) Version 9 diagnosed AMI based on the results of the coronary angiography in Hunan Province, China. Only patients with type 1 AMI, defined as plaque-mediated culprit lesions, were included in our study. The exclusion criteria were age younger than 18 years or older than 90 years; the length of hospital stay <1 day with a lack of information about the entire hospitalization for AMI, including information on death; severe hepatic impairment (alanine transaminase (ALT) > 400 U/L or total bilirubin (Tbil) > 340 μmol/L); malignancy; life-threatening infection; severe renal impairment (estimated glomerular filtration rate [eGFR] < 15 mL/min); severe and active autoimmune disease or haemodynamically unstable trauma at admission. This study was approved by the Ethics Committee of the Third Xiangya Hospital of Central South University and Xiangtan Central Hospital (IRB approval numbers: 2019-S489 and 2020-11-001, respectively).

For the development cohort, 1,723 patients with AMI who were admitted between January 2013 and December 2017 were consecutively recruited for this study from the Third Xiangya Hospital of Central South University in Changsha city. The last date of follow-up for the mortality was August 31, 2021. Patients with severe hepatic dysfunction ($n = 58$), severe renal impairment ($n = 85$), active malignancy ($n = 54$), severe autoimmune disease ($n = 8$), current life-threatening infection ($n = 39$), and haemodynamically unstable trauma ($n = 4$) at admission as well as those with a length of hospital stay <1 day ($n = 4$) were excluded. The remaining 1,471 patients were included in the final analysis of the development cohort. Among the patients in the final analysis, 292 patients died, and nine patients were censored during the follow-up period.

For the external validation cohort, 1,365 patients with ICD-9-confirmed AMI who were admitted between January 2013 and December 2017 were consecutively included from the Xiangtan Central Hospital in Xiangtan city. The last date of the follow-up was August 31, 2021. Patients with severe hepatic dysfunction ($n = 1$), severe renal impairment ($n = 24$), active malignancy ($n = 16$), autoimmune disease ($n = 5$), and haemodynamically unstable trauma ($n = 2$) at admission as well as those with a length of hospital stay <1 day ($n = 66$) were excluded. Thus, 1,251 patients in the external validation cohort were included in the final analysis. Among these patients, 186 patients died, and 65 patients were censored during the follow-up period.

The demographics and clinical characteristics, medical history, laboratory tests, imaging examinations, and in-hospital interventions were obtained from patients' electronic medical records. Mortality and its associated causes were collected from the death registry reporting system of the National Center for Disease Control and Prevention (CDC) and were further confirmed through telephone follow-up. The clinical data for the construction of the predictive models were based on the measurements at admission for both cohorts. The ethics committee of each hospital approved the study protocols. The need to obtain informed consent from the patients was waived by both ethics
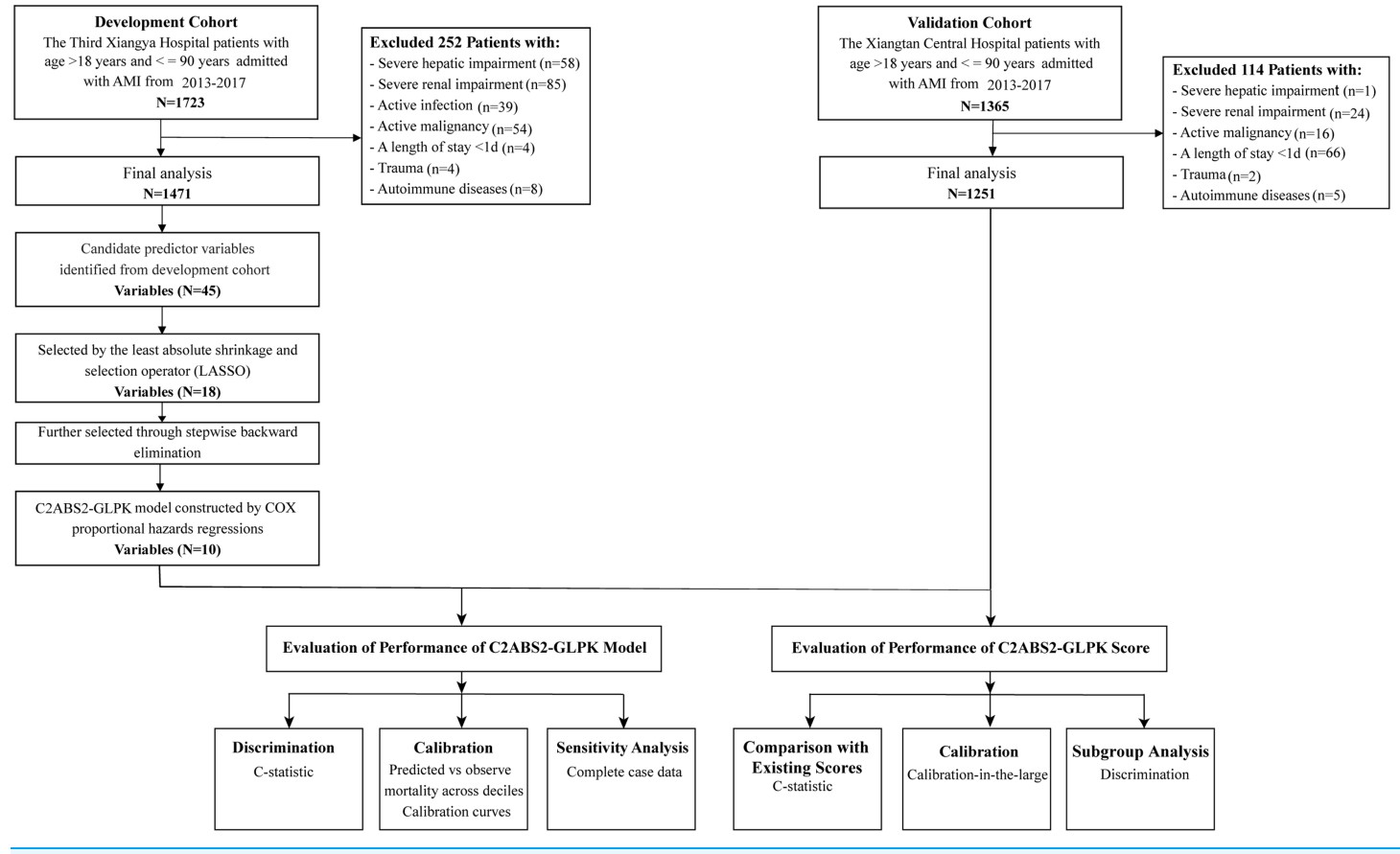

**Figure 1 Flowchart for patient selection and distribution of the development and validation cohorts.** Abbreviations: AMI, acute myocardial infarction.

committees. The study complied with the requirements in the Transparent Reporting of a Multivariable Prediction Model for Individual Prognosis or Diagnosis (TRIPOD) statement (Table S1) (*Moons et al., 2015*). A flowchart of the model development and validation process is shown in Fig. 1.

## Variable selection and model development

Forty-five candidate variables were identified from the clinical variables available at hospital admission and a literature review and entered into the selection process (Table S2). Of these, candidate variables with less than 20% missing data were included in the study (Table S3). The variables selected for the predictive risk model were chosen from a total of the forty-five candidate baseline parameters in a two-round process.

Least absolute shrinkage and selection operator (LASSO) regression was applied in the first-round of variable selection in the development cohort. LASSO regression is capable of minimizing the potential collinearity of variables and overfitting with the minimum lambda ($\lambda = 0.037$), and the mean squared error (MSE) is within one standard error of minimal MSE using ten cross-validations (*Tibshirani, 1997*; *McClelland et al., 2015*). Eighteen variables were selected from the first-round variable selection and entered into the second-round variable selection. To further validate the variable selection, we repeated

the random forest algorithm 100 times in the development cohort. The variables were evaluated by the importance and the out-of-bag (OOB) value (*Kruppa et al., 2014*; *Speiser, Durkalski & Lee, 2015*).

The second round of variable selection was used to further select variables by multivariate stepwise backward regression model. Five iterative processes were applied to develop a parsimonious model while minimizing overfitting. An Akaike information criterion (AIC) value <4.6 and a *p*-value > 0.05 were the cut-off values for the chosen final variables. Ten variables were selected from the second round of variable selection.

In the final model building stage, ten variables selected from the second-round of variable selection were utilized in a multivariate Cox hazards regression model to develop an algorithm for predicting 5-year all-cause mortality in AMI patients. Due to the unknown individual absolute survival rate, we could obtain only the relative risks of AMI patients compared with those without any risk factors based on the regression coefficients of all variables (*Kalbfleisch & Prentice, 2002*; *Royston & Altman, 2013*). To improve the application of the C2ABS2-GLPK score in clinical practice, we transformed the Cox model into a nomogram model. Every selected variable was given a continuous score from 0 to 100 corresponding to the regression coefficient. The total risk score incorporating the ten predictors corresponded to the relative survival rate.

The X-tile program was used to identify the optimal cut-off value of the risk score to stratify patients with various degrees of mortality risk in the development cohort. The optimal cut-off values and patient groups were based on the minimal *p*-value (0.003) and the maximum of chi-square log-rank value (*p*-value < 0.001) according to 5-year mortality in patients with AMI. The optimal cut-off values were also validated in the external validation cohort (log-rank *p*-value < 0.001) (*Camp, Dolled-Filhart & Rimm, 2004*).

X-tile analysis revealed that patients with C2ABS2-GLPK scores ≤148 points were stratified into a low-risk group with a 5-year mortality of 6.6%; Patients with C2ABS2-GLPK scores between 149 and 218 points were graded as medium-risk with a 5-year mortality of 15.2%; and patients with C2ABS2-GLPK scores ≥219 points were categorized into a high-risk group with a 5-year mortality of 37.3%. Moreover, based on our literature review, the predicted mortality in each risk subgroup was similar to that of the GRACE score, thrombolysis in myocardial infarction (TIMI) risk score and modified GRACE score during a follow-up period of more than 5 years (*Fox et al., 2010*; *Hall et al., 2018*; *Kozieradzka et al., 2011*; *Henderson et al., 2015*).

## Evaluation of model performance

The performance of the risk prediction model was evaluated by its discrimination and calibration. Internal validation was performed in the primary development cohort. Given the censoring in the development and external validation cohorts (*Hippisley-Cox, Coupland & Brindle, 2013*; *Kalderstam et al., 2013*), 1,000-times bootstrapped C-statistics were applied to assess the discrimination of the risk prediction model in the development and external validation cohorts.

Calibration measures how accurately a model's predictions match the overall observed mortality in the follow-up period. We equally split the observed 5-year mortality into equal deciles across the development and external validation datasets. Calibration performance was assessed with a calibration plot and summarized across the full range of mortality deciles using the Hosmer-Lemeshow statistic. The calibration plot is generated from the means and standard deviation (SD) of the calibration slope and intercept. Furthermore, calibration of the C2ABS2-GLPK score was measured by calibration-in-the-large, which is capable of reporting the difference between the overall observed 5-year mortality and the mean predicted mortality in the development cohort and external validation cohort.

## Missing data imputation, cross validation and sensitivity analyses

Imputation for missing variables was considered if fewer than 20% of values were missing. Multiple imputations were implemented to handle missing data with the "mice" package, which is a widely accepted method for population datasets (*Azur et al., 2011*). Continuous and categorical data were imputed in a 10 sets with a 10-iteration process based on the fitted conditional models until a stopping criterion was satisfied.

Missing data patterns were analysed by the "finalfit" package, and data were considered to be missing at random, which means that each variable in the dataset is equally and randomly likely to be missing and that the conditional likelihood of a missing value is partly dependent on other missing variables, defined by the observed data. Missing at random is in contrast to missing completely at random (*Knight et al., 2020*).

The linear regression model for continuous variables (mean error rate 15.4%, 95% CI [12.3–16.7%]), logistic regression model for binary variables (mean error rate 6.1%, 95% CI [5.5–7.8%]) and polytomous logistic regression for categorical variables (mean error rate 7.3%, 95% CI [6.5–9.2%]) with more than two levels were applied to the missing parameters to estimate the imputation errors and internally cross-validated errors (*Waljee et al., 2013*; *Knight et al., 2020*).

To address the potential effect of missing data imputation, complete case data excluding missing values were analysed to assess the robustness of the C2ABS2-GLPK model to predict 5-year all-cause mortality. Thus, we performed sensitivity analyses by using complete case data to assess the discrimination of the C2ABS2-GLPK model by C-statistics with 1000 iterations of bootstrapping in the development and external validation cohorts, and to compare the results with those of the imputed datasets.

## Comparison with existing prognostic scores

The performance of the C2ABS2-GLPK score was compared with that of existing prognostic scores, including the GRACE score, KAMIR score and China Acute Myocardial Infarction (CAMI) score, in the development and the external validation cohorts. The discrimination of the GRACE score, KAMIR score and CAMI score was conducted in the samples with 1,000 iterations of bootstrapping for both cohorts. Differences in the C-statistics were assessed using the DeLong test (*Harrell, Lee & Mark, 1996*; *Steyerberg, 2008*; *Steyerberg et al., 2010*). Moreover, the bootstrapped discrimination of the

C2ABS2-GLPK score was also evaluated in the subgroups based on sex, smoking statuses, ST segment elevation myocardial infarction (STEMI), type 2 diabetes, dyslipidaemia (low-density lipoprotein cholesterol [LDL-C] > 2.59 mmol/L), or obesity (body mass index [BMI] > 24) in the development and external validation cohorts.

To evaluate the net benefits of the C2ABS2-GLPK score at the threshold probability for 5-year mortality in the development and external validation sets, we also performed a decision curve analysis to make a clinical judgement about the relative values of mortality (predicting a true positive) and survival rate (predicting a false positive) associated with clinical prediction scores (*Vickers & Elkin, 2006*). The standardized net benefit was plotted against the threshold probability of considering a patient at high risk of 5-year mortality for the C2ABS2-GLPK score alone and the existing GRACE, KAMIR and CAMI scores applicable to more than 50% of the Chinese AMI patients in the external validation cohort.

## Statistical Analysis

Group comparisons were conducted using the chi-squared test for equal proportions or a t tests for normally distributed data; otherwise, the Wilcoxon rank sum tests was used. A $p < 0.05$ was considered indicative of statistical significance. Statistical analysis was performed using R software (version 3.6.2, R Foundation).

## RESULTS

### Patients characteristics in the development cohort

A total of 1,471 patients with ICD-9-confirmed AMI between January 2013 and December 2017 from the Third Xiangya Hospital, Changsha city, Hunan Province, were included in the development cohort. During a median follow-up of 5 years (range: 0.08 to 8.75 years), 292 patients (19.9%) died, and 9 (0.6%) were lost to follow-up before the end of the follow-up period. The median age was 64 years (interquartile range (IQR), 55–72), and 71.4% of the patients were men. A total of 173 (11.8%) patients had a history of stroke, 35 (2.4%) experienced preadmission or at-admission cardiac arrest, 844 (57.4%) underwent immediate percutaneous transluminal coronary intervention [PCI], 1,457 (99.0%) were administered statins at admission, and 87 (5.9%) developed acute heart failure ranked as Killip IV at admission. The median N-terminal pro-brain natriuretic peptide [NT-proBNP] level at admission was 1,639 pg/ml (IQR 611–3009), the median serum creatinine at admission was 82 μmol/L (IQR, 68-99), and the median left ventricular end-diastolic diameter at admission was 49 mm (IQR, 45–51). The baseline characteristics and their differences among the survivors, the deceased patients and the censored patients in the cohort are shown in Table 1.

### Variable selection and the C2ABS2-GLPK model establishment

Forty-five candidate clinical variables at admission from the patients in the development cohort were included for variable selection. Among these variables, 18 variables were selected in the first-round LASSO regression (Fig. S1, Table S4). The variables with an average OOB value >20% obtained by repeating the random forest algorithm were similar

**Table 1 Baseline characteristics of patients with AMI in the model development and validation database.**

| Variables | Development database | | | | Validation database | | | |
|---|---|---|---|---|---|---|---|---|
| | Total (N = 1,471) | Survivors (N = 1,170) | Non-survivors (N = 292) | Censor (N = 9) | Total (N = 1,251) | Survivors (N = 1,000) | Non-survivors (N = 186) | Censor (N = 65) |
| Age, median (IQR), y | 64 (55–72) | 62 (53–70) | 71 (63–77) | 72 (68–76) | 65 (56–73) | 63 (54–71) | 72 (66–78) | 66 (59–72) |
| Male, n (%) | 1,050 (71.4) | 860 (73.5) | 187 (64.0) | 3 (33.3) | 911 (72.8) | 751 (75.2) | 113 (60.8) | 47 (72.3) |
| BMI, median (IQR), kg/m$^2$ | 23.9 (21.8–26.0) | 24.0 (22.0–26.0) | 23.8 (21.4–25.7) | 21.9 (20.7–23.3) | 23.4 (22.2–24.8) | 23.4 (22.2–24.9) | 23.4 (22.0–24.0) | 23.4 (23.4–24.0) |
| Current smoking, n (%) | 156 (10.6) | 126 (10.8) | 29 (9.9) | 1 (11.1) | 80 (6.4) | 67 (6.7) | 10 (5.4) | 3 (4.6) |
| ST segment depression, n (%) | 497 (33.8) | 388 (33.2) | 107 (36.6) | 2 (22.2) | 256 (20.5) | 196 (19.6) | 47 (25.3) | 13 (20.0) |
| STEMI, n (%) | 922 (62.7) | 757 (64.7) | 159 (54.5) | 6 (66.7) | 919 (73.5) | 764 (76.4) | 108 (58.1) | 47 (72.3) |
| Acute Anterior MI, n (%) | 494 (33.5) | 401 (34.3) | 77 (26.4) | 6 (66.7) | 451 (36.1) | 373 (37.3) | 58 (31.2) | 20 (30.8) |
| Left main coronary lesion, n (%) | 107 (7.3) | 85 (7.3) | 22 (7.5) | 0 (0.0) | 114 (9.1) | 89 (8.9) | 21 (11.3) | 4 (6.2) |
| Coronary multivessel lesion, n (%) | 515 (35.0) | 417 (35.6) | 94 (32.2) | 4 (44.4) | 34 (2.7) | 22 (2.2) | 8 (4.3) | 4 (6.2) |
| Door-to-Balloon time, n (%), h | | | | | | | | |
| ≤4 | 191 (13.0) | 146 (12.5) | 43 (14.7) | 2 (22.2) | 391 (31.3) | 315 (31.5) | 65 (34.9) | 11 (16.9) |
| >4 | 1,280 (87.0) | 1024 (87.5) | 249 (85.3) | 7 (77.8) | 857 (68.5) | 701 (70.1) | 111 (59.7) | 45 (69.2) |
| Symptoms, n (%) | | | | | | | | |
| Cardiac Arrest | 35 (2.4) | 11 (0.9) | 24 (8.2) | 0 (0.0) | 38 (3.0) | 15 (1.5) | 20 (10.8) | 3 (4.6) |
| Killip Classification, n (%) | | | | | | | | |
| I | 1,028 (69.9) | 890 (76.1) | 133 (45.5) | 5 (55.6) | 698 (55.8) | 597 (59.7) | 63 (29.8) | 38 (58.5) |
| II | 274 (18.6) | 199 (17.0) | 73 (25.0) | 2 (22.2) | 404 (32.3) | 321 (32.1) | 66 (35.5) | 17 (26.2) |
| III | 82 (5.6) | 43 (3.7) | 37 (12.7) | 2 (22.2) | 83 (6.6) | 48 (4.8) | 29 (15.6) | 6 (9.2) |
| IV | 87 (5.9) | 38 (3.2) | 49 (16.8) | 0 (0.0) | 66 (5.3) | 34 (3.4) | 28 (15.1) | 4 (6.2) |
| Signs, median (IQR) | | | | | | | | |
| HR, beats/min | 75 (66–87) | 74 (66–85) | 80 (70–92) | 86 (76–92) | 78 (68–88) | 78 (68–87) | 80 (70–95) | 80 (73–90) |
| Blood Pressure, median (IQR), mm Hg | | | | | | | | |
| Systolic | 128 (111–142) | 128 (112–142) | 128 (110–143) | 124 (110–126) | 130 (119–149) | 130 (120–150) | 130 (115–146) | 130 (120–143) |
| Diastolic | 76 (68–84) | 76 (69–84) | 76 (66–84) | 76 (74–80) | 80 (70–90) | 80 (70–90) | 80 (70–90) | 80 (75–90) |
| Laboratory Findings, median (IQR) | | | | | | | | |
| WBC, *10$^9$/L | 9.0 (7.1–11.4) | 9.0 (7.2–11.3) | 9.1 (6.8–12.0) | 9.5 (6.3–10.6) | 9.4 (7.5–11.9) | 9.4 (7.5–11.8) | 9.4 (7.5–12.2) | 9.7 (6.8–12.4) |
| Hb, g/L | 130 (117–144) | 133 (120–145) | 122 (109–133) | 121 (109–130) | 130 (118–142) | 131 (119–142) | 127 (116–141) | 133 (120–141) |
| PLT, *10$^9$/L | 203 (166–246) | 205 (169–249) | 194 (155–235) | 164 (113–252) | 189 (153–226) | 190 (153–227) | 189 (155–232) | 188 (149–219) |
| ALT, U/L | 36 (23–54) | 37 (24–54) | 33 (20–57) | 33 (31–40) | 34 (20–49) | 34 (19–48) | 35 (20–52) | 36 (23–51) |
| Cr, μmol/L | 82 (68–99) | 80 (68–94) | 93 (75–126) | 79 (65–101) | 79 (67–95) | 78 (66–93) | 87 (72–112) | 75 (65–82) |
| FBG, mmol/L | 5.9 (5.1–7.2) | 5.9 (5.0–7.1) | 6.4 (5.2–8.1) | 4.5 (4.2–8.3) | 5.9 (5.0–7.3) | 5.9 (5.0–7.1) | 6.2 (5.1–8.1) | 6.3 (5.1–6.9) |
| LDL-C, mmol/L | 2.5 (1.9–3.0) | 2.5 (2.0–3.0) | 2.4 (1.9–2.9) | 2.3 (2.0–2.6) | 2.8 (2.2-3.3) | 2.8 (2.2–3.3) | 2.7 (2.2–3.2) | 3.0 (2.4–3.5) |
| Biomarkers cardiac injury, median (IQR) | | | | | | | | |
| NT-proBNP, pg/ml | 1,639 (611–3,009) | 1,341 (514–2,639) | 3,347 (1,601–6,763) | 5,520 (3,450–6,267) | 918 (213–2,587) | 720 (185–2,587) | 2,587 (828–5,886) | 1,183 (351–2,587) |
| cTnI, μg/L | 5.2 (1.0–15.1) | 5.4 (0.9–15.3) | 4.9 (1.1–13.7) | 1.0 (0.4–2.6) | 1.0 (0.2–4.8) | 0.9 (0.2–4.0) | 1.8 (0.3–6.8) | 0.6 (0.1–3.2) |
| Prior History, n (%) | | | | | | | | |
| Prior MI | 28 (1.9) | 21 (1.8) | 6 (2.1) | 1 (11.1) | 11 (0.9) | 7 (0.7) | 3 (1.6) | 1 (1.5) |

| Table 1 (continued) | | | | | | | | |
|---|---|---|---|---|---|---|---|---|
| Variables | Development database | | | | Validation database | | | |
| | Total (N = 1,471) | Survivors (N = 1,170) | Non-survivors (N = 292) | Censor (N = 9) | Total (N = 1,251) | Survivors (N = 1,000) | Non-survivors (N = 186) | Censor (N = 65) |
| Prior PCI | 22 (1.5) | 16 (1.4) | 5 (1.7) | 1 (11.1) | 90 (7.2) | 71 (7.1) | 9 (4.8) | 10 (15.4) |
| Comorbidities, n (%) | | | | | | | | |
| Hypertension | 828 (56.3) | 640 (54.7) | 182 (62.3) | 6 (66.7) | 703 (56.2) | 574 (57.4) | 112 (60.2) | 39 (60.0) |
| Diabetes | 478 (32.5) | 369 (31.5) | 104 (35.6) | 5 (55.6) | 218 (17.4) | 162 (16.2) | 40 (21.5) | 16 (24.6) |
| Stroke | 173 (11.8) | 111 (9.5) | 60 (20.5) | 2 (22.2) | 127 (10.2) | 97 (9.7) | 25 (13.4) | 5 (7.7) |
| AF | 23 (1.6) | 14 (1.2) | 9 (3.0) | 0 (0.0) | 73 (5.8) | 55 (5.5) | 13 (7.0) | 5 (7.7) |
| Echocardiography | | | | | | | | |
| LVEF, median IQR), % | 54 (49–61) | 54 (50–62) | 54 (44–54) | 46 (30–54) | 51 (44–58) | 51 (45–59) | 50 (39–56) | 50 (44–56) |
| LA, median (IQR), mm | 33 (31–35) | 33 (30–35) | 33 (33–36) | 34 (28–38) | 35 (32–37) | 35 (32–37) | 35 (33–37) | 35 (32–37) |
| LVDd, median (IQR), mm | 49 (45–51) | 49 (45–50) | 49 (48–53) | 49 (45–55) | 49 (46–51) | 49 (45–51) | 49 (47–55) | 49 (46–53) |
| RA, median (IQR), mm | 30 (29–32) | 30 (28–32) | 30 (30–33) | 30 (27–31) | 35 (33–36) | 35 (33–36) | 35 (34–36) | 35 (33–37) |
| RV, median (IQR), mm | 29 (27–31) | 29 (27–31) | 29 (28–31) | 29 (28–30) | 19 (18–19) | 19 (18–19) | 19 (18–19) | 19 (17–19) |
| Aortic Regurgitation, n (%) | 365 (24.8) | 262 (23.0) | 100 (34.2) | 3 (33.3) | 551 (44.1) | 418 (42.2) | 106 (57.0) | 27 (41.5) |
| Mitral Regurgitation, n (%) | 978 (66.5) | 739 (64.0) | 233 (79.8) | 6 (66.7) | 970 (77.5) | 780 (78.0) | 144 (77.4) | 46 (70.8) |
| Tricuspid Regurgitation, n (%) | 555 (37.7) | 433 (37.1) | 118 (40.4) | 4 (44.4) | 910 (72.7) | 722 (72.2) | 144 (77.4) | 44 (67.7) |
| Pulmonary Regurgitation, n (%) | 220 (15.0) | 171 (15.1) | 47 (16.1) | 2 (22.2) | 30 (2.4) | 27 (2.7) | 2 (1.1) | 1 (1.5) |
| Decreased Left Ventricular Compliance, n (%) | 473 (32.2) | 417 (35.3) | 54 (18.5) | 2 (22.2) | 1,144 (91.4) | 916 (91.6) | 170 (91.4) | 58 (89.2) |
| Baseline medications, n (%) | | | | | | | | |
| Antiplatelets | 1461 (99.3) | 1,168 (99.8) | 284 (97.3) | 9 (100.0) | 1,232 (98.5) | 984 (98.4) | 183 (98.4) | 65 (100.0) |
| Statins | 1457 (99.0) | 1,166 (99.6) | 282 (96.6) | 9 (100.0) | 1,251 (100.0) | 1,000 (100.0) | 186 (100.0) | 65 (100.0) |
| Antihypertensives | 1354 (92.1) | 1,106 (94.0) | 239 (81.8) | 9 (100.0) | 1,096 (87.6) | 873 (87.3) | 161 (86.6) | 62 (95.4) |
| PCI | 844 (57.4) | 760 (65.0) | 81 (27.7) | 3 (33.3) | 867 (69.3) | 719 (71.9) | 99 (53.2) | 49 (75.4) |
| CABG | 2 (0.1) | 2 (0.2) | 0 (0.0) | 0 (0.0) | 1 (0.1) | 0 (0.0) | 1 (0.5) | 0 (0.0) |

Note:
AMI, acute myocardial infarction; IQR, interquartile range; BMI, body mass index; STEMI, ST segment elevation myocardial infarction; Door-to-Balloon time: Time from hospital arrival to first balloon inflation; HR, heart rate; WBC, white blood cell; Hb, hemoglobin; PLT, platelet; ALT, alanine transaminase; Cr, creatinine; FBG, fast blood glucose; LDL-C, low-density lipoprotein cholesterol; NT-proBNP, N-terminal pro-brain natriuretic peptide; cTnI, cardiac Troponin I; MI, myocardial infarction; PCI, percutaneous transluminal coronary intervention; AF, atrial fibrillation; LVEF, left ventricular ejection fraction; LA, left atrial; LVDd, left ventricular end-diastolic diameter; RA, right atrial; RV, right ventricular; LV, left ventricular; Antiplatelets, aspirin, clopidogrel, ticagrelor; Antihypertensives, angiotensin-converting enzyme inhibitor, angiotensin receptor blocker, calcium-channel blocker, β-receptor blocker; CABG, coronary artery bypass grafting.

to the variables selected by LASSO regression. The importance of variables, as assessed by the random forest algorithm and reported as the percent increase in MSE (%IncMSE), is shown in Fig. S2 and Table S5. After five rounds of multivariable stepwise backward regression, the 18 LASSO-selected variables were reduced to 10 variables to enter into the final multivariable Cox proportional hazards model (Table S6) and to construct the C2ABS2-GLPK model. The final 10 risk factors, namely, admission or perihospital cardiac arrest, a history of stroke, Killip classification II–IV, fasting blood glucose ≥10 mmol/L, left ventricular end-diastolic diameter (LVDd) ≥60 mm, age ≥70, serum creatinine ≤35 μmol, NT-proBNP ≥300 pg/ml, absence of immediate statin administration or PCI within 24 h after symptom onset, were incorporated into the risk score model for the prediction of 5-year mortality in AMI patients (Table 2).

**Table 2 Selected Variables and Cox Model for Predicting 5-year Morality in the Development Cohort.**

| Variables | Hazard ratio (95% CI) | p value |
|---|---|---|
| **Cardiac Arrest** | | |
| No | 1.0 [reference] | |
| Yes | 5.0 [3.1–8.2] | <0.001 |
| **Stroke** | | |
| No | 1.0 [reference] | |
| Yes | 1.7 [1.2–2.3] | 0.001 |
| **Killip, classifications** | | |
| I | 1.0 [reference] | |
| II | 1.3 [1.0–1.9] | 0.034 |
| III | 2.5 [1.7–3.6] | <0.001 |
| IV | 3.9 [2.7–5.6] | <0.001 |
| **FBG, mmol/L** | | |
| 0–10 | 1.0 [reference] | |
| ≥10 | 1.5 [1.1–2.2] | 0.016 |
| **LVDd, mm** | | |
| 0–60 | 1.0 [reference] | |
| ≥60 | 1.9 [1.3–2.8] | 0.002 |
| **Age, years** | | |
| <40 | 1.0 [reference] | |
| 40–49 | 2.0 [0.3–16.5] | 0.506 |
| 50–59 | 3.1 [0.4–23.5] | 0.268 |
| 60–69 | 6.0 [0.8–44.1] | 0.080 |
| 70–79 | 7.5 [1.0–55.5] | 0.048 |
| 80–89 | 11.1 [1.5–82.9] | 0.019 |
| **Cr, μmol/L** | | |
| 0–35 | 14.5 [1.9–110.4] | 0.010 |
| 36–70 | 1.0 [reference] | |
| 71–105 | 1.2 [0.9–1.6] | 0.289 |
| 106–140 | 1.5 [1.0–2.2] | 0.035 |
| 141–176 | 2.4 [1.5–3.8] | <0.001 |
| ≥177 | 2.3 [1.4–4.1] | 0.002 |
| **NT-ProBNP, pg/ml** | | |
| 0–300 | 1.0 [reference] | |
| ≥300 | 2.0 [1.1–3.5] | 0.024 |
| **Statins therapy** | | |
| No | 1.0 [reference] | |
| Yes | 0.2 [0.1–0.5] | <0.001 |
| **PCI** | | |
| No | 1.0 [reference] | |
| Yes | 0.4 [0.3–0.5] | <0.001 |

**Note:**
CI, confidence intervals; FBG, fast blood glucose; LVDd, left ventricular end-diastolic diameter; Cr, Creatinine; NT-proBNP, N-terminal pro-brain natriuretic peptide; PCI, percutaneous transluminal coronary intervention.

## Construction of the prognostic nomogram and C2ABS2-GLPK score

The nomogram that incorporated the final 10 significant independent factors for predicting the 5-year survival rate in the development cohort was established according to their regression coefficients from the C2ABS2-GLPK model (Fig. 2).

Based on the nomogram, every selected variable was given a continuous score from 0 to 100 corresponding to the regression coefficient. The total C2ABS2-GLPK score was the sum of the scores of each variable. The vertical line was dropped down from the total score row to estimate the relative 1-, 2-, 3-, 4-, and 5-year all-cause survival rates for better application in clinical practice (Fig. 2). The details of the score for each variable are shown in Table S7.

## The performance of the C2ABS2-GLPK model and score in the development cohort and internal validation

In the development cohort, the C2ABS2-GLPK model showed good discrimination in predicting 5-year all-cause mortality in patients with AMI, with a C-statistic equal to 0.811 (95% CI [0.786–0.836]) (Table S8). Using the Hosmer-Lemeshow goodness-of-fit test, we found that this model was well calibrated with good coherence between observed and predicted mortality across deciles with the a Hosmer-Lemeshow statistic index of 5.277 ($p = 0.728$) (Table S8, Fig. 3). The calibration slope was 0.988, and the intercept was 0.716 in the prediction of 5-year mortality using the C2ABS2-GLPK model (Table S9). A slope of approximately 1.0 in the development dataset indicated that the C2ABS2-GLPK model was perfect; however, the intercept was typically far from 0.0, suggesting that the predicted risk was an overestimation of 5-year mortality in AMI patients. Nevertheless, the overall observed mortality *vs* the mean predicted mortality for the C2ABS2-GLPK score were similar (calibration-in-the-large = 0.02) (Fig. 3).

According to the C2ABS2-GLPK score, AMI patients in the development cohort were stratified into three distinct risk groups by the predicted mortality probabilities following the cut-off points detected by the X-tile analysis. The predicted probability of dying within 5 years was 6.6% in the low-risk group (total points ≤148 points), 15.2% in the medium-risk group (total points, 149–218 points), and 37.3% in the high-risk group (total points >219 points). Compared to that in the low-risk group, the HRs (95% CI) for the medium- and high-risk groups were 2.4 (95% CI [1.6–3.6]) and 7.2 (95% CI [5.1–10.2]), respectively (log-rank $p < 0.001$) (Fig. 2, Table S10).

## The performance of the C2ABS2-GLPK model and score in the external validation cohort

A total of 1,251 patients with ICD-9-confirmed AMI between January 2013 and December 2017 from Xiangtan Central Hospital, Xiangtan city, Hunan Province, were included in the final external validation cohort (Table 1). During a median follow-up of 5 years (range: 0.08 to 8.58 years), 186 patients (14.9%) died, and 65 (5.2%) were lost to follow-up. The median age was 65 years (IQR, 56–73), and 72.8% of subjects were men. A total of 127 (10.2%) patients had a history of stroke, 38 (3.0%) experienced preadmission and at-admission cardiac arrest, 867 (69.3%) underwent immediate PCI, 1,251 (100.0%) were

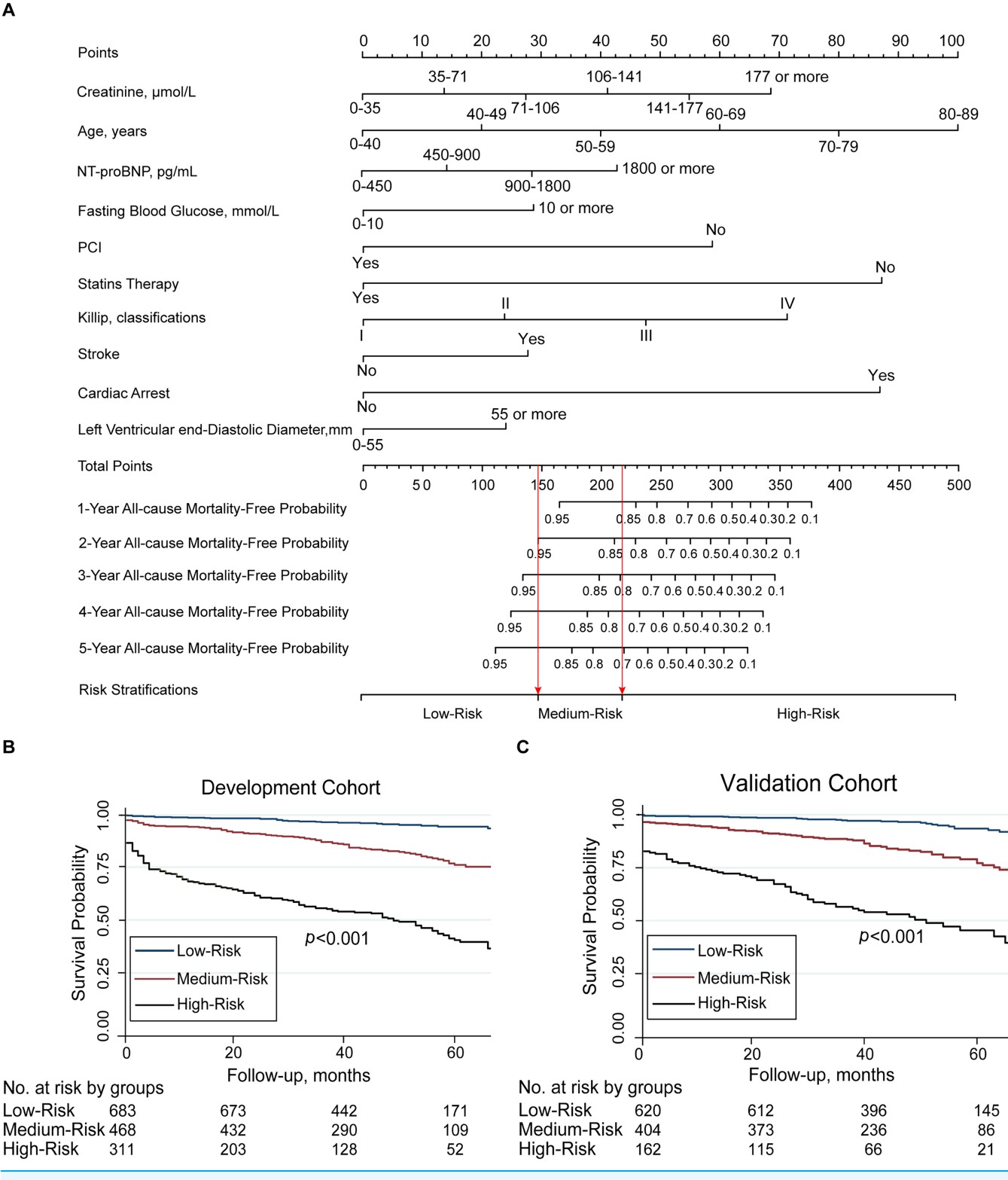

**Figure 2 Nomogram for C2ABS2-GLPK score to predict 1-, 2-, 3-, 4-, 5-year all-cause mortality in patients with AMI and Kaplan-Meier survival curves stratified by C2ABS2-GLPK score.** (A) Nomogram for predicting 1-, 2-, 3-, 4-, 5-year all-cause mortality among patients with AMI. (B) Kaplan–Meier survival curves for AMI patients with risk stratified by C2ABS2-GLPK score according to the X-tile analysis in the development cohort. (C) Kaplan–Meier survival curves for ami patients with risk stratified by C2ABS2-GLPK score according to the X-tile analysis

![PeerJ]

**Figure 2** (continued)
in the external validation cohort. Notes: Nomogram for the prediction of 1-, 2-, 3-, 4-, 5-year all-cause mortality-free in patients with AMI. The patient's age is located on the row labeled "Age, y", the patient's left ventricular end-diastolic diameter is located on the row labeled "Left Ventricular end-diastolic Diameter, mm", the patient's fasting blood glucose level is located on the row labeled "Fasting Blood Glucose, mmol/L", the patient's creatinine level is located on the row labeled "Creatinine, μmol/L", the patient's NT-proBNP level is located on the row labeled "NT-proBNP, pg/ml", the patient's Killip classification is located on the row labeled "Killip, classifications", and a straight line is drawn up to the row labeled "Points" to determine the corresponding points. This process is repeated for each of the remaining factors by drawing a straight line to the "Points" row to determine the points associated with each factor. After summing the total points, one locates the appropriate total point number and draws a straight line from this to the rows labeled Predicted 1-, 2-, 3-, 4-, 5-Year All-cause Mortality-Free probability to determine the patient's predicted survival probability. Abbreviations: AMI, acute myocardial infarction; NT-proBNP, N-terminal pro-brain natriuretic peptide; PCI, percutaneous transluminal coronary intervention.               

administered statins, and 66 (5.3%) developed acute heart failure ranked as Killip IV at admission. The median NT-proBNP level was 918 pg/ml (IQR, 213–2,587), the median serum creatinine was 79 μmol/L (IQR, 67–95), and the median left ventricular end-diastolic diameter at admission was 49 mm (IQR, 46–51). The C-statistic of the C2ABS2-GLPK score, indicating its discrimination ability, was 0.787 (95% CI, 0.756–0.818) in predicting 5-year all-cause mortality for Chinese AMI patients in the external validation cohort (Table S8). Calibration was also found to be excellent in the external validation cohort: the Hosmer-Lemeshow statistic index for comparison of the observed and predicted risk across deciles of 5-year all-cause mortality in the C2ABS2-GLPK model was 9.495 ($p = 0.302$) (Fig. 3, Table S8). The slope was 0.976, and the intercept was 0.473 in the prediction of 5-year mortality using the C2ABS2-GLPK model (Table S9). Although a slope nearly equal to 1.0 in the external validation dataset suggested that the C2ABS2-GLPK model was good, the intercept was well above 0.0, indicating that the predicted risk was an overestimation of 5-year mortality in AMI patients. Calibration was also found to be excellent for the C2ABS2-GLPK score: the overall observed mortality *vs* the mean predicted mortality was nearly equal (calibration-in-the-large = 0.03) (Fig. 3).

The C2ABS2-GLPK score was also applied to the external validation cohort. In the validation cohort, AMI patients could still be divided into the same three risk categories according to the total prediction score. The mean observed 5-year mortality rates in the low-, medium-, and high-risk groups were 8.8%, 19.6%, and 41.7%, respectively. The Kaplan–Meier curves showed that the HRs (95% CIs) for the medium-risk and high-risk categories were 2.6 (1.8–3.6) and 6.5 (4.6–9.1), respectively, compared to that of the low-risk category (log-rank $p < 0.001$) (Fig. 2, Table S10). The results indicated that a higher C2ABS2-GLPK score was associated with an increased risk of 5-year mortality ($p < 0.001$).

## Comparing the performance of C2ABS2-GLPK score with existing scores

To further validate the superiority in assessing the performance of C2ABS2-GLPK score, we used the c-statistics to compare this score to the GRACE score, KAMIR score and CAMI score in the development cohort and external validation cohort, respectively. The ability of each score was assessed by c-statistic (95% CI). Among them, the c-statistic

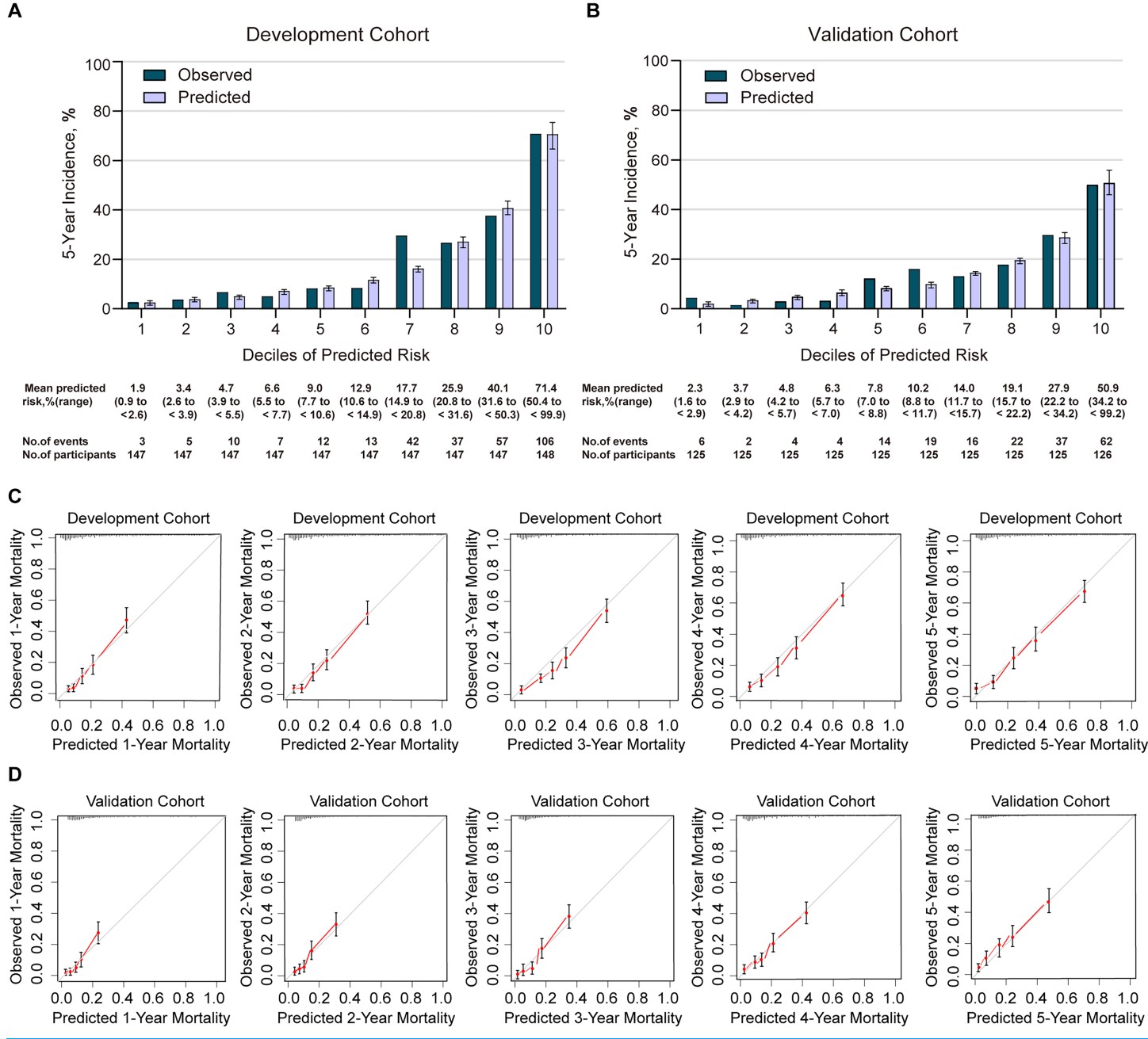

**Figure 3 Calibration of C2ABS2-GLPK score in the development cohort and validation cohort.** (A) Agreement between the deciles of observed vs predicted 5-year mortality in the cox model of the development cohort. (B) Agreement between the deciles of observed *vs* predicted 5-year mortality in the cox model of the validation cohort. (C) The calibration curves for predicting mortality from the first year to the fifth year of follow-up in the development cohort. (D) The calibration curves for predicting mortality from the first year to the fifth year of follow-up in the validation cohort. Notes: Predicted 5-year mortality for each decile is the mean predicted risk score in each decile. Error bars indicate standard deviations.                               

for C2ABS2-GLPK score either in the development cohort (c-statistic, 0.811; 95% CI [0.786–0.836]) or the external validation cohort (c-statistic, 0.787; 95% CI [0.756–0.818]) were all significantly higher than those of the GRACE score (c-statistic, 0.728; 95% CI [0.697–0.759]) (*p* < 0.001), KAMIR score (c-statistic, 0.783; 95% CI [0.758–0.808])

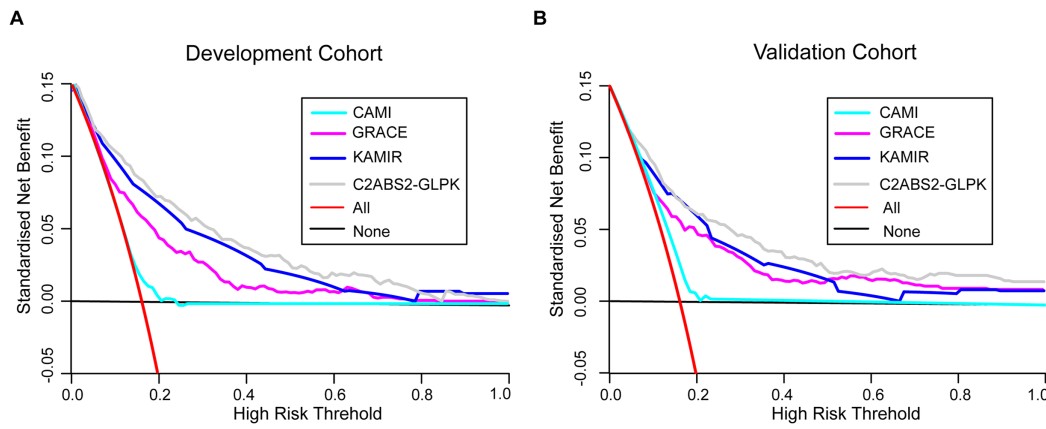

**Figure 4 Decision curve analysis for C2ABS2-GLPK, GRACE, KAMIR and CAMI scores.** (A) Decision curve analysis for C2ABS2-GLPK, GRACE, KAMIR and CAMI scores in the development cohort. (B) Decision curve analysis for C2ABS2-GLPK, GRACE, KAMIR and CAMI Scores in the external validation cohorts. Notes: Decision curve analysis for most discriminating four scores applicable to more than 50% of validation population (restricted cubic spline; imputed cohorts). Lines are shown for standardised net benefit at different risk thresholds of treating no patients (black line) and treating all patients (red line). Abbreviations: GRACE, Global Registry of Acute Coronary Events; KAMIR, Korea Acute Myocardial Infarction Registry; CAMI: China Acute Myocardial Infarction.

($p < 0.001$) and CAMI score (c-statistic, 0.558; 95% CI [0.523–0.593]) ($p < 0.001$), which reflects the higher prognostic value of the C2ABS2-GLPK score for the prediction of 5-year mortality in Chinese patients with AMI (Table S8).

To further assess the clinical utility of the C2ABS2-GLPK score, we also performed decision curve analyses to compare this score with the GRACE score, KAMIR score and CAMI score. The results showed that the C2ABS2-GLPK score had better clinical utility across the threshold of 5-year all-cause mortality applicable to more than 50% of both the development cohort and the external validation cohort (Fig. 4).

## Sensitivity analysis

To further confirm the reliability of the C2ABS2-GLPK model, we performed sensitivity analyses by using complete case data. The results showed that the discrimination with the C-statistic based on complete case data was 0.805 (95% CI [0.774–0.836]) in the development cohort and 0.780 (95% CI [0.743–0.817]) in the external validation cohort, which was similar to that with C-statistics of 0.811 (95% CI [0.786–0.836]) and 0.787 (95% CI [0.756–0.818]) in the imputed development and validation datasets, respectively (Table S11).

## Subgroup validation

Consistent with the performance in the whole cohort, the predictive performance of the C2ABS2-GLPK score was robust across all subgroups, with C-statistics ranging from 0.770 to 0.832 in subgroups of patients according to sex, history of smoking, types of AMI, and comorbidities of diabetes, obesity and hypercholesterolaemia in the internal and external validation cohorts. Furthermore, the performance of the C2ABS2-GLPK score

was superior to that of the GRACE score (c-statistics range: 0.684 to 0.765), the KAMIR score (c-statistics range: 0.744 to 0.807), and the CAMI score (c-statistics range: 0.477 to 0.580) in the indicated subgroups (Table S12, Fig. S3).

## DISCUSSION

In this study, we developed and validated a clinical prognostic score (C2ABS2-GLPK score) assessed within 24 h of admission to predict 5-year mortality among Chinese patients with AMI in two independent cohorts, which allows for the potential collinearity of independent predictors and overfittings (*Tibshirani, 1997*; *McClelland et al., 2015*). Overall, the C2ABS2-GLPK score had satisfactory discrimination and calibration performance for the 5-year prediction of all-cause mortality in Chinese patients with AMI. With reference to the GRACE scoring system, the patients in the medium-risk group with a C2ABS2-GLPK score between 149 and 218 might be suggested to have guideline-indicated pharmacotherapies and evidence-based care. Patients with a C2ABS2-GLPK score above 219 in the high-risk group of C2ABS2-GLPK score might be recommended to have 'up to standard' guideline-indicated care, intensified therapy, and additional lifestyle modifications and follow-up to improve their long-term survival (*Hall et al., 2018*). The performance and risk stratification abilities of the C2ABS2-GLPK score warrant further validation in prospective studies.

Over the past 30 years, some risk scores for predicting long-term mortality in patients with AMI, such as the GRACE (*Kozieradzka et al., 2011*), KAMIR (*Kim et al., 2011*), SAMI (*Plakht et al., 2012*), PAMI (*Addala et al., 2004*), TIMI (*Kozieradzka et al., 2011*) and Zwolle scores (*Kozieradzka et al., 2011*), have been developed and validated. However, these scores were based on Western and Korean populations due to differences in medical interventions, responses to therapy, economic levels and genetic backgrounds across countries; consequently, they may not be well suited to predict long-term mortality for Chinese AMI patients. Recently, the China AMI (CAMI) registry, a nationwide registry of hospitalized patients with AMI, was designed and launched in China. Although there are plans for predicting 2-year mortality of AMI in the CAMI database, the CAMI score is currently focused only on in-hospital mortality (*Xu et al., 2016*; *Fu et al., 2018*; *Tang et al., 2019*). Additionally, the CAMI score might not be easily accessible in the clinic due to its two scoring systems specific to CAMI-STEMI and CAMI-NSTEMI. Therefore, we developed the C2ABS2-GLPK score for predicting long-term mortality in Chinese AMI patients to be simple and easily available in clinical practice. We also validated that the C2ABS2-GLPK score outperformed the GRACE, KAMIR and CAMI scores in estimating the 5-year risk of death in AMI patients. In addition, the C2ABS2-GLPK score, which was established using clinical parameters and medical intervention at admission, may guide the optimization of interventional strategies under emergency circumstances.

Compared to the widely used classic GRACE scoring system, our C2ABS2-GLPK score might be somewhat superior in the following aspects. **First**, it is well known that the GRACE score for predicting more than 6-month mortality in AMI patients is assessed at hospital discharge (*Eagle et al., 2004*). However, the C2ABS2-GLPK score is calculated within 24 h of admission and might be more timely for optimizing the therapeutic and

monitoring regimen during hospitalization to eventually improve long-term survival for AMI patients in China. **Second**, the GRACE risk model was developed and validated based on data from 1999 to 2003 (*Allen et al., 2007*). With advances in the treatment of AMI, the GRACE score may not be suitable for current AMI patients. Our C2ABS2-GLPK score integrated advanced interventional procedures and medication to fill in the gaps of the current scores in predicting long-term mortality for AMI patients. **Third**, compared with the GRACE score, the C2ABS2-GLPK score in the present study increased the accuracy of predicting long-term prognosis by adding a new dimension, the status of target organs such as the heart and brain, which has been proven to predict long-term mortality in patients with AMI (*Israr et al., 2018*; *Richards et al., 2003*; *Ndrepepa et al., 2006*; *Taylor et al., 2007*; *Brammås et al., 2013*). **Fourth**, in comparison to the GRACE score, the C2ABS2-GLPK score places more emphasis on the essential role of statins in the early treatment of AMI, which has recently been considered to be an independent predictor of one-year major adverse cardiovascular events in patients with AMI (*Kim et al., 2019*).

The C2ABS2-GLPK score provided a comprehensive evaluation of cardiac function from clinical manifestation (Killip classification), serum biochemical indicators (NT-proBNP), and structure of the cardiac chamber(cardiac ultrasound) in the acute phase. The Killip classification primarily considers cardiac findings during physical examination, which has proven to be essential in predicting mortality in AMI patients in dozens of studies (*Khot et al., 2003*; *Stebbins et al., 2010*; *Fu et al., 2018*; *Song et al., 2018*; *Chen, Han & Luo, 2019*; *Meyer et al., 2019*; *Wang et al., 2021*). However, this classification may not be sensitive to detecting asymptomatic heart failure. NT-proBNP, a sensitive biomarker for heart failure, is secreted from the atria and ventricles and is increased in both symptomatic and asymptomatic patients with cardiac dysfunction. Furthermore, cardiac ultrasound adds a definitive diagnosis and evaluation of the structure and function of each cardiac chamber, valve, and attachments. Therefore, integration of the three indicators improved both the sensitivity and specificity of the evaluation of cardiac function.

The PLATelet inhibition and patient Outcomes (PLATO) trial reported that ACS patients with a history of stroke or transient ischaemic attack (TIA) had higher rates of 1-year death (10.5%) than those without a history of stroke or TIA (4.9%) (*Mahaffey et al., 2014*). A systematic review showed that AMI patients with a history of stroke had a higher risk of mortality than patients without stroke at the one-year follow-up (*Johansson et al., 2017*). Our study also identified that prior stroke or acute stroke at admission is an important predictor of 5-year mortality in patients with AMI.

Numerous studies have shown that statins benefit in-hospital and long-term survival in AMI patients *via* multiple mechanisms, such as lipid reduction, anti-inflammation and plaque stabilization (*Heart Protection Study Collaborative Group, 2002*; *Merx et al., 2005*; *Sim et al., 2013*). Existing evidence from a Chilean registry showed that early initiation of statins improved in-hospital survival in patients with AMI (*Martínez et al., 2013*). Concordant with these findings, we study revealed that early statin therapy improved long-term survival in patients with AMI.

Cardiac arrest has been well demonstrated to substantially increase in-hospital and 30-day mortality risk (*Newby et al., 1998*; *Fordyce et al., 2016*). However, studies have shown that AMI patients with perihospital cardiac arrest did not have higher 1-year mortality than those without prehospitalization cardiac arrest (*Lee et al., 2014*; *Fordyce et al., 2016*). Our study revealed that cardiac arrest is the strongest predictor for long-term mortality in AMI patients.

There are some limitations of our study. **First,** the modest sample size used to construct the risk score and validation may influence the accuracy of the model. Future studies with a larger population are warranted. **Second,** the cohorts for score development and validation are from one province in China, which could limit the generalizability of the score to patients in other areas of China. Additional validation studies in patients with AMI from areas outside Hunan Province is worth performing in further studies. **Third,** this model was based on the retrospective development and validation cohorts, and prospective studies may further increase the reliability of this model. **Fourth,** 45 candidate variables were selected and used to predict 5-year mortality. There is a potential risk of overfitting during score development; **Fifth,** a retrospective design cannot rule out the influence of unmeasured confounders, such as economic status and compliance, that may impact long-term mortality. **Sixth,** since we selected the candidate variables based on a literature review, other potential factors impacting long-term mortality in AMI patients were possibly excluded during the initial variable selection. Inevitable selection bias may exist and lead to the overestimation of risk. Future studies are warranted to confirm the robustness of our score.

## CONCLUSIONS

In summary, the C2ABS2-GLPK score was established to estimate 5-year mortality in AMI patients in China based on ten variables that are commonly measured at hospital admission. The C2ABS2-GLPK score showed good performance in predicting long-term outcomes in AMI patients, however, warrant to be further validation in additional cohorts and in prospective settings is warranted. This tool may help physicians identify patients who are at high risk of long-term mortality, thus optimizing medical treatments and eventually improving long-term survival of AMI patients in China.

### Funding
This work was supported by Key Research and Development Plan of Hunan Province (No. S2021GCZDYF1348); the National Natural Science Foundation of China (No. 81870171; No. 81800393; No. 81770403; No. 81974054); the National Key Research and Development Projects (No. 2019YFF0216300; No. 2019YFF0216305); the National Key Research and Development Projects (No. 2018YFC1311300). There was no additional external funding received for this study. The funders had no role in study design, data collection and analysis, decision to publish, or preparation of the manuscript.

## Grant Disclosures

The following grant information was disclosed by the authors:

Key Research and Development Plan of Hunan Province: S2021GCZDYF1348.

National Natural Science Foundation of China: 81870171, 81800393, 81770403, 81974054.

National Key Research and Development Projects: 2019YFF0216300, 019YFF0216305.

National Key Research and Development Projects: 2018YFC1311300.

## Competing Interests

The authors declare that they have no competing interests.

## Author Contributions

- Yan Tang conceived and designed the experiments, performed the experiments, analyzed the data, prepared figures and/or tables, authored or reviewed drafts of the paper, and approved the final draft.
- Yuanyuan Bai performed the experiments, analyzed the data, prepared figures and/or tables, authored or reviewed drafts of the paper, and approved the final draft.
- Yuanyuan Chen performed the experiments, analyzed the data, authored or reviewed drafts of the paper, and approved the final draft.
- Xuejing Sun analyzed the data, authored or reviewed drafts of the paper, and approved the final draft.
- Yunmin Shi analyzed the data, authored or reviewed drafts of the paper, and approved the final draft.
- Tian He analyzed the data, authored or reviewed drafts of the paper, and approved the final draft.
- Mengqing Jiang analyzed the data, authored or reviewed drafts of the paper, and approved the final draft.
- Yujie Wang analyzed the data, authored or reviewed drafts of the paper, and approved the final draft.
- Mingxing Wu analyzed the data, authored or reviewed drafts of the paper, and approved the final draft.
- Zhiliu Peng analyzed the data, authored or reviewed drafts of the paper, and approved the final draft.
- Suzhen Liu analyzed the data, authored or reviewed drafts of the paper, and approved the final draft.
- Weihong Jiang analyzed the data, authored or reviewed drafts of the paper, and approved the final draft.
- Yao Lu analyzed the data, authored or reviewed drafts of the paper, and approved the final draft.
- Hong Yuan analyzed the data, authored or reviewed drafts of the paper, and approved the final draft.
- Jingjing Cai conceived and designed the experiments, analyzed the data, authored or reviewed drafts of the paper, and approved the final draft.

## Human Ethics

The following information was supplied relating to ethical approvals (*i.e.*, approving body and any reference numbers):

The Third Xiangya Hospital of Central South University granted Ethical approval to carry out the study within its facilities (Ethical Application Ref: 2019-S489); The Xiangtan Central Hospital granted Ethical approval to carry out the study within its facilities (Ethical Application Ref: 2020-11-001).

## Data Availability

The raw measurements are available in the Supplementary Files.

## Supplemental Information

Supplemental information for this article can be found online at http://dx.doi.org/10.7717/peerj.12652#supplemental-information.

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
