# Peer review of "Development and validation of a novel risk score to predict 5-year mortality in patients with acute myocardial infarction in China: a retrospective study"

_PeerJ, doi:10.7717/peerj.12652_

## Round 0.1 · original submission · Major Revisions

The reviewers have found scientific merit in your work, but there are some issues that you should address in a revised version of the text. Please, see their comments so as to have more information.

Reviewer 1 ·

Basic reporting

1.The details of missing value for each variables should be complemented in a separate table.
2.The polishing of the language and presentation of the manuscript is necessary.

Experimental design

1.The author needs to explain whether the sample size is sufficient and how to calculate the sample size.
2.Whether the enrolled patients included both type 1 and type 2 AMI. In view of the significant differences in the pathogenesis between the two types, it may not be appropriate to use a universal model.
3. In the final model, it may not be appropriate to directly include the continuous variable, but the continuous variable can be transformed into the classification variable into the final model. eg, if the glucose was included in the model as a continuous variable, whether it meant that the higher glucose was, the higher the mortality rate was, and the lowerthe glucose was, the lower the mortality was. It is obviously unreasonable.

Validity of the findings

CAMI score was designed to predict the prognosis of Chinese patients with AMI as stated by the author in the Discussion part. The comparison of prediction performance between C2ABS2-GLPK and CAMI scores need to be conducted, if possible.

Additional comments

The present study aimed to construct a prediction nomogram to predict the 5-year all-cause mortality of AMI. The subjects were hospitalized between January 2013 and December 2017, and the follow-up was completed in December 2019, which meant we didn't know exactly if the most of subjects hospitalized after 2015 died five years later. The follow-up can be continued in the future.

Reviewer 2 ·

Basic reporting

The article is well constructed and the study flowchart is nice and clear, however English writing can be improved.
There are A LOT of figures and tables and I'm not sure all are needed or provide interpretable information. Please review what you would like to include and their relevance to the findings in the manuscript.

The risk score is reported as a continuous number going up into the hundreds, yet in Figure 4 it is also referred to as a % predicted risk. How was an absolute risk obtained from a Cox model that provides relative risks?
Also, the three final risk bands that are produced include a huge range of risk in the intermediate risk group (4.3-26.5% 5-year risk of death). Please comment on how useful the risk bands are likely to be in clinical practice.

Calibration is assessed, which is good. Please show calibration plots in terms of cumulative probability, not survival. With the current calibration plots, the groups are all squashed into the low risk/high survival end of the scale, and we miss seeing just how accurate risk is predicted in the relevant risk range. Also, the production of calibration plots mean the risk has been converted to absolute risk, so how come the score is reported in the unhelpful scale that goes up into the hundreds?
Calibration is quantified with the calibration slope and intercept, and thank you for including the intercept. It is often overlooked but is vital to show the accuracy of the risk estimates. Here, intercepts were typically far from 0, with values of 0.2-0.4, showing likely overestimation of risk, yet it is not mentioned in the interpretation of the results. Please include.

Why are both the c-statistic and the AUROC calculated? And reported interchangeably?

Hazard ratios don't need to be reported to 3 decimal places (see Table 2). Given the small cohort and low number of events, I suggest 1dp would be appropriate.

Experimental design

Patients were excluded from the study cohort if their length of stay was <1 day. The number excluded on this basis were low but please clarify how many of these were due to the patient dying?

The risk score aims to predict risk of death over 5 years. How many people had the opportunity for 5 years of follow-up?

One of the rationales for developing this score is that a score is needed to predict death during the in-patient stay as well as in the longer period. Yet it is only ever tested in the medium to longer term. I couldn't see where it's performance was assessed for in-hospital or convalescent phase post-MI. Please clarify exactly when the score is intended to be used and the results of the assessment that support such use.

A variable in the final model was "PCI (immediate or delayed)". How delayed? This would also impact on when the score can be used. (as it can't be applied if it needs information about something that happens in the future).

Validity of the findings

The study followed the TRIPOD guidance for reporting of prognostic models, which is great to see. Derivation and validation cohorts were clearly defined and used from the outset, and sensitivity analyses were clearly reported. It was also good that the authors compared the new score to a selection of existing scores for this patient group. The new score derived from the type of cohort used to test it in will always have an advantage over completely independent scores but there was still a point to be made over the better discrimination of the new score over GRACE, at least.

Importantly, the limitations of the study are well described.

There was a lot of variable selection which I'd question is statistically appropriate. However, the goal of a prognostic model is to predict risk, as opposed to the etiological or statistical importance of individual predictors, so an allowance for that can be made. And in terms of performance, the ability of the new score to discriminate, or rank, risk seems well supported. Calibration, or accuracy, of risk appears ok but we can't really assess it given the format of the calibration plots and information that is currently provided.

---

## Round 0.2 · accepted · Accept

All the reviewers' concerns have been correctly addressed. Therefore, I recommend the publicaiton of this work in PeerJ. Congratulations!